**Data Availability Statement:** All relevant data are within the manuscript and its Supporting information files.

**Funding:** Breast Cancer Research Foundation (https://www.bcrf.org) (BCRF-18-086) (ARK) and

# Chaperonin containing TCP1 as a marker for identification of circulating tumor cells in blood

Amanda Cox[1], Ana Martini[1], Heba Ghozlan[1], Rebecca Moroose[2], Xiang Zhu[1], Eunkyung Lee[3], Amr S. Khaled[4], Louis Barr[5], Carlos Alemany[6], Na'im Fanaian[7], Elizabeth Griffith[8], Ryan Sause[9], S. A. Litherland[10], Annette R. Khaled[1]*

1 Burnett School of Biomedical Science, College of Medicine, University of Central Florida, Orlando, Florida, United States of America, 2 Orlando Health Cancer Institute, Orlando, Florida, United States of America, 3 Department of Health Science, College of Health Professions and Sciences, University of Central Florida, Orlando, Florida, United States of America, 4 Pathology and Laboratory Medicine, Orlando VA Medical Center, Orlando, Florida, United States of America, 5 AdventHealth Cancer Institute-Surgical Oncology, Orlando, Florida, United States of America, 6 AdventHealth Cancer Institute-Oncology/Hematology, Clinical Research, Orlando, Florida, United States of America, 7 Central Florida Pathology Associates, Orlando, Florida, United States of America, 8 AdventHealth Cancer Institute-Clinical Research, Orlando, Florida, United States of America, 9 AdventHealth Heart of Florida-Pathology, Orlando, Florida, United States of America, 10 AdventHealth Cancer Institute-Translational Research, Orlando, Florida, United States of America

* annette.khaled@ucf.edu

## Abstract

Herein we report the use of Chaperonin-Containing TCP-1 (CCT or TRiC) as a marker to detect circulating tumor cells (CTCs) that are shed from tumors during oncogenesis. Most detection methods used in liquid biopsy approaches for enumeration of CTCs from blood, employ epithelial markers like cytokeratin (CK). However, such markers provide little information on the potential of these shed tumor cells, which are normally short-lived, to seed metastatic sites. To identify a marker that could go beyond enumeration and provide actionable data on CTCs, we evaluated CCT. CCT is a protein-folding complex composed of eight subunits. Previously, we found that expression of the second subunit (CCT2 or CCTβ) inversely correlated with cancer patient survival and was essential for tumorigenesis in mice, driving tumor-promoting processes like proliferation and anchorage-independent growth. In this study, we examined CCT2 expression in cancer compared to normal tissues and found statistically significant increases in tumors. Because not all blood samples from cancer patients contain detectable CTCs, we used the approach of spiking a known number of cancer cells into blood from healthy donors to test a liquid biopsy approach using CCT2 to distinguish rare cancer cells from the large number of non-cancer cells in blood. Using a clinically validated method for capturing CTCs, we evaluated detection of intracellular CCT2 staining for visualization of breast cancer and small cell lung (SCLC) cancer cells. We demonstrated that CCT2 staining could be incorporated into a CTC capture and staining protocol, providing biologically relevant information to improve detection of cancer cells shed in blood. These results were confirmed with a pilot study of blood from SCLC patients. Our studies demonstrate that detection of CCT2 could identify rare cancer cells in blood and has

The Cathy Engelman Cancer Research
Collaborative Fund (ARK). The funders and
sponsors did not play a role in the study design,
data collection and analysis, decision to publish, or
preparation of the manuscript.

**Competing interests:** We have read the journal's
policy and one of the authors of this manuscript
(Dr. Annette Khaled) has the following competing
interests: [shareholder in Seva Therapeutics, Inc.]
This commercial entity holds a license to use
intellectual property developed by the inventor (Dr.
Khaled) and provided no funding and had no role in
the design, preparation, or submission of this
manuscript and did not employ any of the authors.
This competing interest does not alter our
adherence to PLOS ONE policies on sharing data
and materials.

application in liquid biopsy approaches to enhance the use of minimally invasive methods for cancer diagnosis.

## Introduction

The demand for protein-folding increases during oncogenesis to support the unregulated growth and survival of cancer cells [1–3]. The molecular chaperone, CCT, interacts with numerous oncoproteins, suggesting that multiple cancer signaling pathways could converge through this chaperonin [4]. Unlike the ubiquitous heat shock proteins, CCT is a multi-sub-unit complex consisting of a double-ringed barrel formed with eight subunits (CCT1-8); proteins fold within an inner chamber in an ATP-dependent fashion [5]. While CCT could fold up to 10% of the normal cell proteome, it may have a heighten role during oncogenesis, supporting the growth of cancer cells through client proteins such cytoskeletal proteins (actin, tubulin), cell cycle mediators (cell division cycle protein 20 (cdc20), cyclin E, p53, etc.) [6, 7], and growth/survival factors (Signal transducer and activator of transcription 3 (STAT3), Myelocytomatosis (MYC), etc.) [4, 8, 9]. Data from our lab and others revealed that the CCT2 sub-unit, or CCTβ, is linked to cancer progression and increased stage/aggressiveness of breast, lung, prostate, hepatocellular, gallbladder, and colonic cancer [10–18]. Additionally, we found that the level of CCT2 was independent of hormone receptor status in breast cancer [12], suggesting that *CCT2* expression could be upstream of tumor lineage-defining markers. Overexpressing *CCT2* in luminal breast cancer cells increased cell proliferation, spheroid growth, and anchorage-independent growth [19], while *CCT2* depletion prevented tumor growth in a syngeneic model of triple-negative breast cancer (TNBC) [13]. Moreover, *CCT2* expression is associated with chemoresistance, metastasis, and epithelial to mesenchymal transition (EMT) markers [20, 21]. This suggests that CCT could enable oncogenic changes as CTCs are shed from tumors to seed distal sites, resulting in metastasis. Therefore, assessing CCT2 protein levels in cancer cells/tissues could provide clinicians with information on tumor progression and metastatic potential.

The diagnosis of solid tumors is typically reached through tissue biopsy. However, new diagnostic methods that are non-invasive and provide a better representation of the whole tumor would improve patient management and monitoring. In recent years, new methods for 'liquid biopsy' have developed but few have yet to gain approval for clinical use by the U.S. Food and Drug Administration (FDA). Liquid biopsy is defined as a non-invasive body fluid collection of tumor information like CTCs or circulating tumor DNA (ctDNA) [22]. For diagnostic purposes, CTCs are of interest since these are shed from the active edges of tumors into the peripheral circulation and typically have short half-lives of 1.0–2.4 hours [23]. CTCs, therefore, provide current information about a patient's tumor status [24, 25]. Importantly, CTCs contain the genetic information that could be used to classify patient tumors [26, 27]. Given this, the promising diagnostic potential of CTCs remains to be fully exploited. Most CTCs detection methodologies only provide information with the goal of enumeration. The gold standard for clinical circulating tumor cell (CTC) enumeration is the FDA-approved Cell-Search® System (CSS). CSS uses the epithelial markers, epithelial cellular adhesion molecule (EpCAM) and pan-CK 8, 18, and 19, for capture, enrichment, and staining of CTCs. The use of the CSS enabled the establishment of prognostic thresholds for the detection of CTCs in breast, prostate, and colon cancers [24, 25, 28]. Yet, liquid biopsy methods for the detection of CTCs are used in clinical settings as a complementary diagnostic to the mainstream tissue

biopsy methods. Reasons include the lack of standardization and validation needed to implement liquid biopsies as routine in the clinic. Another reason is tumor heterogeneity, even within the same patient, and the lack of biomarkers for CTC detection to overcome this hurdle.

In this study, to advance the use of non-invasive liquid biopsy for monitoring cancer patients, we investigated the use of CCT2 as a marker for detection of rare cancer cells in blood. The use of CCT2 for the detection of CTCs is supported by our data showing that CCT2 protein levels are increased in tumor tissues compared to normal tissues and may be a better marker for metastatic tissues than CK. The addition of an anti-CCT2 antibody to the CSS platform resulted in improved CTC image analysis and increased detection of rare breast cancer and SCLC cells spiked into blood, which was confirmed in a pilot study of blood from SCLC patients. The detection of CK⁻/CCT2⁺ cancer cells with the CSS platform suggested that the use of CCT2 as a marker could help establish prognostic thresholds for cancers that currently lack validated liquid biopsy methods. Because CCT2 may drive biological processes that are independent of cancer lineage markers and increases with advanced cancer stage, CCT2 could be a novel marker for detection of CTCs that helps enumerate as well as inform on tumor metastatic potential.

## Materials and methods

### Collection of data from online bioinformatics databases

Data was collected from University of California Santa Cruz (UCSC) Xena at xena.ucsc.edu using the combined cohort of "The Cancer Genome Atlas (TCGA), Therapeutically Applicable Research to Generate Effective Treatments (TARGET), and Genotype Tissue Expression (GTEx) samples" dataset [29]. From the available samples, 17,200 from TCGA and GTEX were used for analysis. The initial analysis examined genetic expression of *CCT1-CCT8* using RNA-Seq by Expectation-Maximization (RSEM) expected count (DESeq2 standardized) expression between normal tissue (GTEx) and primary cancer (TCGA) in overall cancer, and then in brain, breast, colon, and lung cancer separately. Secondary analysis of UCSC Xena focused on the expression of the *CCT2* and the CK genes Keratin 8 (*KRT8*), Keratin 18 (*KRT18*), and Keratin 19 (*KRT19*). Phenotype/sample types that were included were: "Metastatic", "Normal tissue", "Primary tumor", and "Solid tissue normal". This data was analyzed through R software for dataset mean and standard deviation (SD). Data for the KMplots was collected at kmplot.com using an mRNA gene chip for breast cancer. Genes used were *CCT2* (201946_s_at) and "mean expression of multiple genes" for CK (*KRT8*: 209008_x_at; *KRT18*: 201596_x_at; and *KRT19*: 201650_at). Hazard ratios and Log-rank p values were calculated by kmplot.com software. mRNAseq data of breast cancer cell lines from the Broad Institute Cancer Cell Encyclopedia (CCLE) were collected by searching for E-cadherin (*CDH1*), N-cadherin (*CDH2*), *EpCAM*, and vimentin (*VIM*) expression in breast cancer.

### Collection of patient blood samples

For the metastatic breast cancer (MBC) study, participants included 38 female patients aged 18 years or older, diagnosed with MBC arising from a prior stage 1, 2, or 3 of disease with paraffin-embedded formalin fixed (FFPE) tissue block available to produce slides for histology. Patients with *de novo* MBC were excluded. Our study size met the parameters established in a sample size estimation based a range of Pearson correlation (R) using the wp.correlation function in the R package (WebPower). If R is $> = 0.5$ (medium effect), then CCT2 expression in CTCs is as clinically relevant as in tumor tissue for prognosis prediction. To detect such a level of effect size (power 0.8, alpha 0.05), we would need 28 patients. If R is $> = 0.6$ (medium-large

effect), we would need 18 patients. The study was conducted observing the human subject protection criteria for the Orlando Regional Medical Center (ORMC) and the University of Central Florida (UCF) and was approved by separate Institutional Review Board (IRB) committees at ORMC and UCF. Informed consent agreements were obtained from all participating subjects. Patients received standard treatments and follow-up at ORMC for breast cancer, at which time two 10 mL of blood in CellSave tubes (Menarini) were drawn and four slides of tissue were obtained from the pathology department at ORMC. Deidentified samples and data on diagnosis, treatment, and recurrence were acquired and processed by the lab at UCF following the Federal Privacy Regulations for protected health information. For the SCLC study, deidentified patient blood from four participants, 7.5–10 mLs of blood in CellSave tubes, was acquired from a commercial source (BioIVT).

## Protocol for standard CTC capture and enumeration

Blood samples in CellSave vials (Menarini) were stored at room temperature and run in CSS according to the manufacturer's protocol for the CSS CTC Kit (Menarini) for CTC analysis within 96 hours of blood draw. CTCs were selected based on the standard CSS criteria for CTC, which is met if the cell is 1) positive for DAPI (4′,6-diamidino-2-phenylindole) staining, 2) positive for CK-FLU (Cytokeratin-fluorescein-conjugated (green)) staining and the CK-FLU stain overlaps with 50% of the DAPI stain, and 3) negative for cluster of differentiation 45 (CD45) staining.

## Immunohistochemistry staining of tissue slides

Slides with FFPE tissues were received from three sites. 1) US Biomax Inc: normal breast tissue microarray (TMA) BRN801b. 2) ORMC: MBC patients from the pilot study described above. 3) AdventHealth: surgical breast cancer archival tissues TMA (>10 yrs) from 28 patients with G1-G3 invasive ductal carcinoma. Slides were processed by standard immunohistochemistry (IHC) methods for CCT2 and STAT3 staining [11]. Anti-CCTβ antibody [amino acids 277 and 473 of Human TCP1 beta] (LifeSpan Biosciences) and STAT3 antibody [E121-21] (Abcam) were used. Slides were stained and scored for the CCT2 stain by an independent and identification-blinded pathologist, as described previously [12]. Images were taken using the BZ-X800 Keyence.

## Cell lines and culture conditions

MDA-MB-231 cells (ATCC® HTB-26™) were cultured in Dulbecco's Modified Eagle's Medium (DMEM) (Corning) and supplemented with 10% fetal bovine serum (FBS) (Gemini) and 1% Penicillin-Streptomycin (P/S) (Corning). T47D cells (ATCC® HTB-133™) were transfected with a lentiviral plasmid to express CCT2 with a FLAG tag (DYKDDDDK) hereafter referred to as T47D-CCT2, as previously described [13]. T47D cells were cultured in Roswell Park Memorial Institute Medium (RPMI-1640) (Corning) supplemented with 10% FBS (Gemini), 1% P/S (Corning), and 0.2 units/mL human recombinant insulin (Santa Cruz). 0.5 μg/mL puromycin dihydrochloride (ThermoFisher) was added to maintain plasmids.

## Surface markers and intracellular staining analysis by flow cytometry

Cells were collected, stained, and analyzed by flow cytometry (BD Cytoflex S flow cytometer) as follows. Staining for membrane surface molecules included use of the following antibodies: EpCAM-PE (phycoerythrin): [VU-1D9] (Abcam) and isotype control [MOPC-21] (BD Pharmingen); E-cadherin-PE: CD324 [DECMA-1] BioLegend) and isotype control [A95-1] (BD

Pharmingen); and N-cadherin-APC (Allophycocyanin): CD325 [8C11] (BioLegend) and iso-type control [27–35] (BD Pharmingen). Staining was performed following standard methods. Cells were stained for 30 min (EpCAM) and 20 min (E-cadherin and N-cadherin). For intra-cellular staining of CCT2 we used the antibody CCT2-PE: [NP_006422] (LSBio) and the iso-type control [MOPC-21] (LSBio), following the method in ThermoFisher Scientific's "Protocol A: two-step protocol: intracellular (cytoplasmic) proteins" and incubating the anti-body for 70 min. When optimizing the CCT2 intracellular stain for CSS Autoprep conditions, we adjusted the antibody staining protocol for CCT2-PE to match the 20 min in the CSS Autoprep automated conditions. All data was generated using FCS Express 6 software (De Novo).

## Reverse transcription-polymerase chain reaction (RT-PCR) for gene expression of EMT markers

RNA was isolated from cells using TRIzol™ (Invitrogen) following the manufacturer's standard protocol for RNA extraction. RNA was quantified using NanoDrop 8000 (ThermoFisher). cDNA was synthesized using the iScript™ cDNA Synthesis Kit (Bio-Rad) following the manu-facturer's protocol. cDNA was diluted 1:5 and mixed with a Fast SYBR™ Green Master Mix (Applied Biosystems) according to manufacturer's recommendations and then run in the Quantstudio PCR for 40 cycles at 95˚C for 3 sec, and 62˚C for 30 sec. Primers were designed using the NCBI nucleotide database, S1 Table.

## Buffer/Manual CSS experiments for optimization of CCT2 antibody in CSS

The protocol was adapted from Lowes et al [30]. In brief, for enrichment CSS reagents from the CSS CXC kit (Menarini) were used: dilution buffer (450 μL), anti-EpCAM ferrofluid (25 μL), and capture enhancement reagent (25 μL). Samples were incubated for 15 min and then added to a magnet holder and incubated for an additional 10 min. The supernatant was removed and discarded. Samples were then stained with the following CSS CXC kit (Menarini) reagents: nucleic acid dye (50 μL), staining reagent (51.5 μL), and permeabilization reagent (100 μL). Anti-CCT2-PE (5 μL) (LSBio: LS-C649415) was added and the sample was incubated for 20 min. After incubation, dilution buffer (500 μL) was added, and the sample was placed in a magnet holder for an additional 10 min. The supernatant was discarded. Samples were re-suspended in dilution buffer (350 μL) and transferred into CSS magnest cartridges and run in CSS Analyzer II. Using the CSS Analyzer II's "Setup" tab, the magnest cartridges could be reconfigured with the "format sample" button. The magnest cartridge could then be labeled for detection with the CXC kit algorithm and the exposure time could be adjusted as described in Lowes et al [30].

## Preparation of blood specimens with spiked cancer cells

Healthy human blood was collected in CellSave vials (Menarini) by a commercial vendor (BioIVT) and shipped overnight. For breast cancer, T47D-CCT2 and MDA-MB-231 cells were spiked into blood at concentration of 1,000 cells per 10 mL of blood. For lung cancer, CRL 5903 and CRL 5853 cells were spiked in blood at a concentration of 100 or 1000 cells per 7.5 mL of blood. Cancer cells were collected from tissue culture conditions in a minimal volume of complete media at the desired concentration and then spiked into the CellSave blood vials. After overnight incubation at room temperature, the samples were run in the CSS using the CSS CXC kit according to the staining protocol for CCT2 listed below. Viability of cells after addition to CellSave vials, under blood-free conditions, was measured using Trypan blue (Gibco) according to the manufacturer's protocol.

## Incorporating the CCT2 stain in the CTC detection protocol and selection of CCT2 positive cancer cells

To incorporate the anti-CCT2-PE antibody into the CSS protocol, we used the Veridex PDF "Guidelines for user-defined markers in CSS" [31]. The PDF can be found at https://www. cellsearchruo.com/sites/default/files/Guideline-for-Use-and-Optimization-of-User-Defined-Markers.pdf. For blood containing spiked cancer cells, anti-CCT2-PE antibody (LSBio) was added at 8 μg/mL using the Veridex PDF recommendations for the staining protocol associated with the CSS CXC kit (Menarini). Lower concentrations (1 μg/mL—4 μg/mL) were also tested with SCLC cell lines. For analysis, the spiked cancer cells were reviewed first using the CSS definition for CTCs which is based on using CK in the Fluorescein isothiocyanate (FITC) channel for identification, and second by the positive expression for CCT2 which is read in the PE channel. Criteria for the CCT2 positive analysis were as follows: 1) round positive signal in the PE channel, 2) PE signal that shows overlap with the DAPI signal, and 3) PE signal that did not show overlap with a CD45 signal if a CD45 signal was present. Cells with faint signal in CD45 channel that could be explained by bleed from the PE channel were not counted as CD45 positive. Cells with pixelated images or streaks in the PE channel were selected if there was a definitive outline between the stained cell and the background. Cells with punctate staining in the PE channel were selected only when the cell outline had a definitive difference from the background. Overlap with CK signal was not a requirement for CCT2 positive cell selection. For the pilot study of SCLC patients, samples were collected and processed according to the "Protocol for standard CTC capture and enumeration" mentioned above. Anti-CCT2 antibody was added at either 12 μg/mL or 24 μg/mL. The same criteria for selection of CTC and CCT2 positive CTC were used as mentioned previously in this section.

## Post-spiked cancer cell processing for detection of cancer cell markers

To assess the presence of cancer markers on breast cancer cells that were enriched after processing in the CSS, cancer cells were recovered after processing and analyzed by flow cytometry (BD Cytoflex S flow cytometer). The CSS CXC kit (Menarini) was used in the CSS Autoprep with the adaptation that the DAPI reagent was removed before the CSS Autoprep cycle began and replaced with PBS. After recovering cells from the CSS magnest, the cells were stained for CD44 with BV421 mouse anti-human CD44 [G44-26] (BD Horizon) or the isotype control [KPL-1] (BD Horizon). The positive control was MDA-MB-231 cells alone (not in blood). The negative control was blood cells alone (not spiked with MDA-MB-231 cells). After staining for 20 min, the samples were analyzed in the BD Cytoflex S and data generated using FCS Express 6 software (De Novo).

## Statistical analysis

StudentT-test and one-way ANOVA statistical analysis was performed to compare gene expression differences between tumor and normal tissue, and among different cancer types. R software was used to calculate mean and SD for UCSC Xena-downloaded data. For RT-PCR analysis, relative gene expression was calculated by dividing a gene's expression in that sample by the average expression of this gene in control samples, i.e., $2^{-\Delta\Delta Ct}$/average ($2^{-\Delta\Delta Ct \text{ control sample}}$) with T47D-lentiviral control (T47D-GFP) assigned as the control sample. A Log-rank test was used to compare cancer patient survival rates between high and low expression for *CCT2* and *CK*. To investigate the association of CCT2 protein levels with CTC and active disease (time gap), Spearman's correlation analysis and multiple linear regression analysis were performed. For regression analysis, we incorporated time gap as a dependent variable and CCT2

as an independent variable, and controlled for CTCs, age, estrogen receptor (ER), progesterone receptor (PR), and human epidermal growth factor receptor 2 (HER2) status. All statistical analyses were performed using Stata MP 15 (StataCorp LLC, 2019) or GraphPad Prism 9 software. All statistical tests were two-tailed with a significance level of α (type I error) <0.05.

## Results

### Bioinformatics-based evaluations of CCT2 or CK/KRT gene expression in normal compared to cancer tissue

To evaluate *CCT* gene expression in tumor tissues compared to normal tissues, the eight CCT subunits (*CCT1-8*) were analyzed in the UCSC Xena Browser using the cohort of "TCGA, TARGET, and GTEx samples" dataset [29]. All eight CCT subunits showed significant (p< 0.0001) upregulation in RNAseq expression in cancer compared to normal tissue, with the strongest upregulation seen in *CCT2* and *CCT3*, S1A Fig. These results were consistent when broken down by cancer type; brain, breast, colon, and lung, S1B–S1E Fig. CK is a standard diagnostic marker for epithelial cancers and CK8, CK18, and CK19 are used in the CSS when visualizing for CTCs. Therefore, to investigate *CCT2* and *CK/KRT* gene expression correlations with tumors and normal tissue, we used the UCSC Xena dataset to focus on *CCT2, KRT8, KRT18,* and *KRT19*. UCSC Xena separates normal tissue from healthy patients (GTEx dataset) and normal tissue from cancer patients (TCGA dataset) by calling them "normal tissue" and "solid tissue normal" respectively. The pair-wise comparison showed that the average *CCT2* gene expression in primary tumor and metastatic tissue was significantly higher than in normal tissue and solid tissue normal (mean±SD: 12.42±0.76 and 12.51±0.72 compared to 11.51±0.62 and 11.86±0.42 respectively, p<0.0001 for all pairwise comparisons). Meanwhile, *KRT8* and *KRT19* gene expression in metastatic tissue showed significant downregulation compared to normal tissue from health patients (mean±SD: 8.51±2.78 vs 9.68±3.51, p<0.0001, and 4.19±3.77 vs 8.47±3.53, p<0.0001) for *KRT8* and *KRT19* respectively, Fig 1A. CCT2 also demonstrated the lowest variance of the four markers. Full statistical analysis is shown in S2 Table. To determine how *CCT2* and *CK/KRT* correlated with overall survival (OS), we used KMplotter to assess these relationships for breast cancer, Fig 1B. Results showed statistically significant inverse correlations between CCT2 high and OS in breast cancer (p = 6.4e-05). High *KRT8/18/19* expression also demonstrated an inverse correlation with the patient OS but was not statistically significant and only the upper quartile survival could be calculated because both cohorts did not reach a median survival. These findings support that *CCT2* expression is upregulated in cancer and metastatic tissues compared to normal tissue, while *KRT8/19* showed downregulation in metastatic tissue, and that *CCT2* expression correlated inversely with patient OS.

### Detection of CCT2 in tumor tissues using breast cancer patient cohorts

To confirm the trend of high *CCT2* gene expression correlating with tumor progression observed in the bioinformatics data, we examined the prevalence of the CCT2 protein in normal breast tissue compared to breast cancer tissues. We stained a breast tissue microarray (TMABRN801b) containing 70 patient samples of normal, adjacent normal, and cancer-adjacent tissue (CAT) breast tissues, and ten samples of invasive ductal carcinoma (IDC) for CCT2. Normal tissue was from breast tissue in patients without cancer, while adjacent normal tissue was from normal breast tissue that was distal to the patient's cancerous tissue. CAT was from breast tissue that was next to the cancerous tissue or organ tissue next to metastatic sites (e.g., liver, bone). We previously reported a CCT2 tissue staining scale from 0 to 4 that was created by a pathologist based on light and dark cytoplasmic and nuclear staining [12]. In our

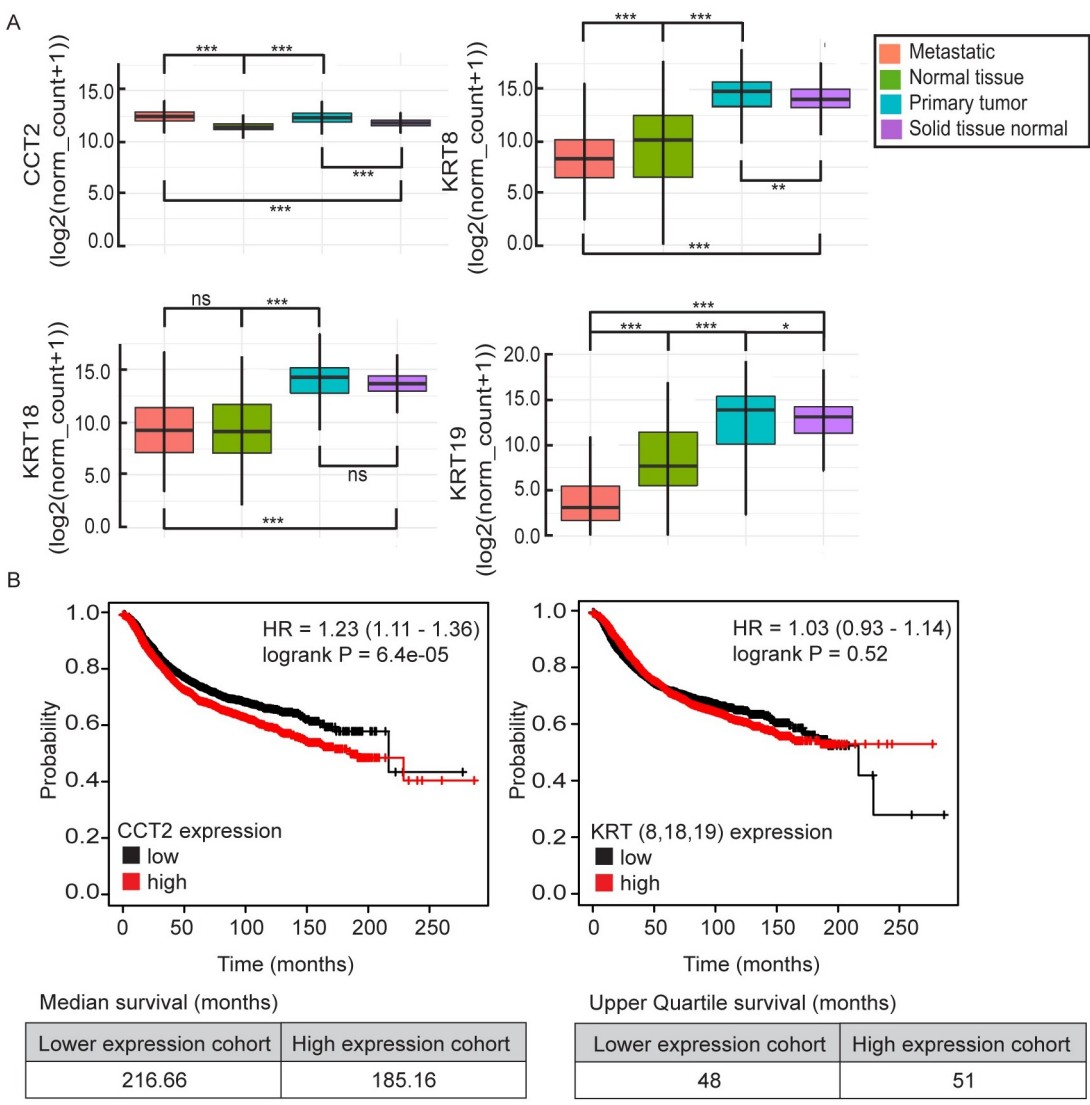

**Fig 1. Bioinformatic and histological analysis of CCT2 in cancer patients.** (A) UCSC Xena analysis from "TCGA, TARGET, GTEx" dataset (n = 17,200) comparing mRNA expression of *CCT2*, *KRT8*, *KRT18*, and *KRT19* in metastatic tissue (orange), normal tissues/GTEx (green), primary tumor (blue), and solid tissue normal/TCGA (purple). * = p<0.05, ** = p<0.005, *** = p<0.0001. (B) KMplotter analysis of overall survival (OS) with low vs. high *CCT2* mRNA expression and low vs high mRNA expression of the mean of *KRT8*, *KRT18*, and *KRT19* combined in breast cancer patients; n = 4,929. Hazard ratios (HR) and log-rank p-values, as calculated by kmplot.com software, are listed on the KmPlots.

tissue data, we present results as CCT2$^{neg}$, (score of 0), CCT2$^{lo}$ (score of 1 or 2), and CCT2$^{hi}$ staining (score of 3 or 4), S2A Fig. Normal breast tissues mostly showed minimal to low CCT2 staining, Fig 2A. Early-stage IDC tumor samples also showed minimal to low CCT2 staining. We next ran a pilot study to evaluate CCT2 protein levels in different stages of breast cancer. A TMA was created using archived patient specimens from ductal and lobular breast cancers of various grades. This TMA was stained for CCT2 and one of its client proteins, STAT3 (representative images of array, S2B and S2C Fig) and contained 13 CCT2$^{lo}$ specimens and 11 CCT2$^{hi}$ specimens that correlated with the corresponding levels of STAT3 staining, S3 Table. Moreover, CCT2$^{hi}$ staining positively correlated with higher cancer stage in both ductal and lobular carcinoma which is consistent with data previously reported by our lab [12].

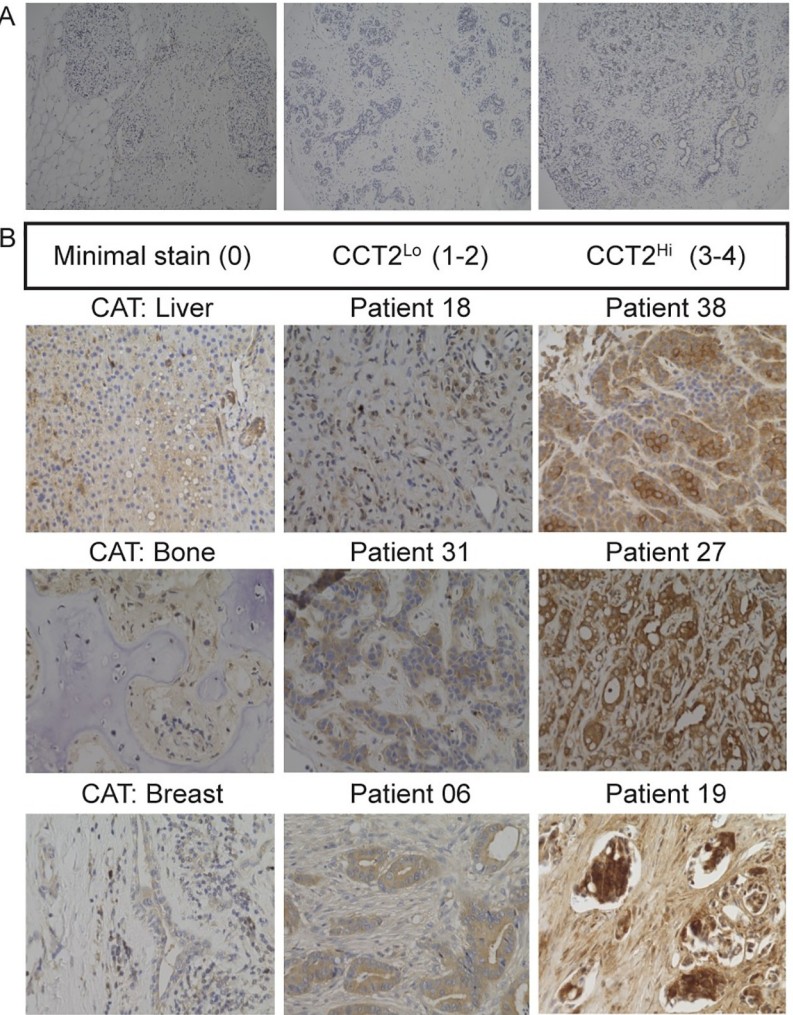

**Fig 2. Histological analysis of CCT2 in normal tissue vs. metastatic breast cancer (MBC) patients.** (A) Representative images from tissue microarray of normal breast tissue stained for CCT2. (B) Representative images from a cohort of MBC patient tissues (breast and metastatic sites; S4 Table) stained for CCT2. The images are—Left column: cancer adjacent tissue (CAT) from locations indicated in the figure, which had minimal staining. Middle column: CCT2$^{lo}$ tissue, which was classified as a score of 1 or 2. Right column: CCT2$^{hi}$ tissue, which was classified as a score of 3 or 4.

The levels of CCT2 in MBC by immunostaining tumor tissues were examined with 38 MBC patients, S4 Table. Fifteen specimens were excluded due to failed screening (n = 4) or inadequate samples (n = 11). The tumor source (primary or metastatic) is listed in the S4 Table. CCT2 score (CCT2$^{lo}$ or CCT2$^{hi}$) is listed in Table 1 and representative images are shown in Fig 2B. In the MBC study, all tumor tissue samples were positive for CCT2 staining. Fourteen of the patients had tumor tissue with CCT2$^{lo}$ staining (score 1 or 2) and 9 of the patients had tumor tissue with CCT2$^{hi}$ staining (score 3 or 4).

### Evaluation of CCT2 tumor score and CTCs enumerated in a cohort of MBC patients

To determine whether tumor levels of CCT2 correlated with prognostic markers for breast cancer, we examined the MBC cohort (Table 1) for levels of CCT2 in the tumor tissue, tumor

**Table 1. Analysis of different prognostic markers from metastatic breast cancer (MBC) patients.**

| Patient Number | Number of CTCs in Blood | CCT2 Score: Low (1–2), High (3–4) | ER Status | PR Status | HER2 Status | Patient Age | Number of yrs. since biopsy "Time Gap" | Number of yrs. since metastatic diagnosis |
|---|---|---|---|---|---|---|---|---|
| 020 | 1 | High | Y | N | N | 61 | 0 | 0 |
| 008 | 2 | Low | Y | Y | N | 75 | 1 | 1 |
| 012 | >5 | Low | N | N | N | 69 | 1 | 1 |
| 013 | 5 | High | Y | N | N | 71 | 1 | 11 |
| 027 | 1 | High | Y | Y | N | 48 | 1 | 1 |
| 032 | 0 | High | Y | Y | N | 51 | 1 | 1 |
| 036 | 2 | Low | N | N | Y | 40 | 1 | 1 |
| 037 | 4 | High | Y | Y | N | 57 | 1 | 0 |
| 038 | 0 | High | Y | Y | N | 70 | 1 | 1 |
| 034 | >5 | High | Y | Y | N | 62 | 2 | 1 |
| 014 | 2 | Low | N | N | N | 63 | 3 | 3 |
| 021 | 0 | Low | Y | Y | N | 57 | 3 | 3 |
| 030 | 0 | Low | Y | Y | N | 80 | 3 | 3 |
| 031 | 0 | Low | N | N | Y | 69 | 3 | 3 |
| 033 | 2 | High | Y | Y | N | 53 | 3 | 1 |
| 017 | 1 | Low | Y | Y | N | 55 | 4 | 0 |
| 019 | 1 | High | Y | Y | N | 68 | 4 | 4 |
| 023 | 4 | Low | Y | Y | N | 66 | 4 | 4 |
| 035 | 1 | Low | Y | Y | N | 50 | 5 | 0 |
| 025 | 0 | Low | NG | NG | NG | 64 | 7 | 1 |
| 018 | 1 | Low | Y | Y | N | 55 | 8 | 3 |
| 028 | 3 | Low | Y | N | Y | 33 | 9 | 4 |
| 006 | 0 | Low | Y | N | Y | 70 | 10 | 4 |

Samples 005, 007, 009, and 010 failed the screening. Samples 001, 002, 003, 004, 011, 015, 016, 022, 024, 026, and 029 were inadequate for either CTC processing or CCT2 staining. Grey highlights indicate patient specimens with five years or more time gap. The "time gap" is defined as the number of years between blood draw and a patient's tumor biopsy. Abbreviations: NG, information not given. Y, yes. N, no.

lineage markers, time of diagnosis, and the number of CTCs enumerated using standard epithelial markers. We collected blood samples from 23 of the MBC patients (S4 Table) and ran CTC analysis according to the FDA-approved standard CSS protocol, Table 1. Although all 23 patients were diagnosed with MBC, only three met the CTC threshold established by CSS for poor prognosis (>5 CTC/7.5 mL of blood) [24]. To determine how the CCT2 tumor score compared to CTC counts with patient outcomes, we ran a Spearman correlation analysis. As a surrogate for active disease, we used the time gap between the date of the most recent tumor tissue biopsy and the date of our blood draw for CTC analysis, Table 1. A larger time gap could indicate a less active disease state, while a shorter time gap could indicate increased patient monitoring and a more active disease state. The result of the Spearman correlation showed that the CCT2$^{hi}$ tumor score had a statistically significant (p = 0.007) inverse correlation with the time gap, Table 2. A multiple linear regression analysis was completed to determine the impact of CCT2 tumor score on time gap when accounting for other variables: age, CTCs, ER, PR, and HER2 status, Table 3. The CCT2 tumor score demonstrated statistically significant (p = 0.009) inverse correlations with the time gap. In comparison, ER status demonstrated statistically significant (p = 0.010) positive correlations with the time gap, which would be expected since this marker is linked to treatments. Additional trends were noted from the data in Table 1. For example, a CCT2$^{lo}$ score correlated with a longer period since the metastatic

**Table 2. Spearman correlation for CCT2 tumor stain score, circulating tumor cell (CTC) count, and time gap.**

|  | CTC | CCT2 tumor stain Score |
|---|---|---|
| **CCT2 Tumor stain Score** | 0.138(p = 0.531) | NA |
| **Time gap** | -0.232(p = 0.287) | *-0.545(p = 0.007)* |

Abbreviation: NA, not applicable.

diagnosis. Also, three out of the five patients with 4–5 CTCs had CCT2[hi] tumor scores. When CCT2[hi] tumor scores did not correlate with high CTC counts, it could be due to differences in the quality of the tumor biopsy as well as treatment effects or it could be from undercounted CTCs due to the reliance on epithelial markers. Results from this MBC cohort provided insight on the relationship between tumor-based CCT2 expression and shed epithelial CTCs in breast cancer, suggesting that CCT2 may perform better as a marker for CTCs than standard epithelial markers alone, which is further explored.

## Optimization of CCT2 staining in the CSS

A limitation with standard CTC detection methods is the reliance on epithelial markers. Such methods could miss cancer cells with hybrid (mixed epithelial and mesenchymal) or mesenchymal features. To include cells in our study that also had mesenchymal markers, we used breast cancer cell lines in which we had previously evaluated CCT2 levels and then determined their expression of epithelial and mesenchymal markers. T47D cells, representative of early-stage luminal A breast cancer, have lower protein levels of CCT2 compared to TNBC cells, like MDA-MB-231 [12, 13, 19]. In our published study of the role of CCT2 in the proliferation of cancer cells, we generated T47D cells that expressed exogenous CCT2 tagged with FLAG (T47D-CCT2) [13]. We assessed these T472-CCT2 cells for *SNAI1 (SNAIL)*, *TWIST*, *EpCAM*, *vimentin*, *E-cadherin*, and *N-cadherin* mRNA expression, Fig 3A. SNAIL and TWIST are transcription factors that are associated with the activation of EMT. EpCAM and E-cadherin are epithelial markers, while vimentin and N-cadherin are mesenchymal markers. Statistically significant increases in *TWIST* (p = 0.0023), *N-cadherin* (p = 0.0170), and *E-cadherin* (p = 0.0215) mRNA were present in the T47D-CCT2 cells compared to the lentiviral control cells, with a significant decrease in *vimentin* (p = 0.0194) mRNA. *EpCAM* levels decreased, and *SNAIL* was increased, but neither of these were statistically significant. Therefore, since T47D-CCT2 cells expressed some mesenchymal characteristics without complete loss of epithelial ones, we used these cells for subsequent experiments. According to the online database CCLE, RNAseq expression of EMT markers indicated that T47D cells are more epithelial-like, and TNBC MDA-MB-231 breast cancer cells are more mesenchymal-like (S3 Fig); hence, we also used MDA-MB-231 cells in our experiments. We examined the membrane protein levels of EpCAM, E-cadherin, and N-cadherin by flow cytometry for both cell lines. T47D-CCT2 cells had detectable levels of EpCAM and E-cadherin and lower N-cadherin, Fig 3B. MDA-MB-231 were N-cadherin positive with a slightly lower expression of EpCAM and E-cadherin than T47D-CCT2, Fig 3C. In total, these data demonstrate that these two cell lines represent a range of epithelial and mesenchymal features.

To validate the anti-CCT2 antibody for intracellular staining, various antibody concentrations were tested in MDA-MB-231 cells and T47D-CCT2 cells to differentiate specific and non-specific signals under the conditions similar to those occurring during automated CTC collection and staining with the CSS. Saturation of the response was achieved around 20 μg/mL of antibody, S4 Fig. To optimize the cell-based signal for CCT2 in the CSS Analyzer II, we

**Table 3. Multiple linear regression for CCT2 tumor stain score and time gap, accounting for various variables.**

|  | Coefficient | SE | 95% C.I. (low) | 95% C.I. (high) | P-value |
|---|---|---|---|---|---|
| **CTC** | 0.005 | 0.251 | -0.530 | 0.539 | 0.986 |
| **CCT2** | -3.063 | 1.026 | -5.251 | -0.876 | *0.009* |
| **ER** | 4.505 | 1.524 | 1.256 | 7.753 | *0.010* |
| **PR** | -0.669 | 1.462 | -3.784 | 2.447 | 0.654 |
| **HER2** | 2.734 | 1.734 | -0.963 | 6.430 | 0.136 |
| **Age** | -0.019 | 0.042 | -0.108 | 0.070 | 0.657 |

Abbreviations: SE: standard error; C.I. confidence interval.

manually performed the intracellular staining for CCT2 in cells [30] and adjusted the exposure time in the CSS Analyzer II and the anti-CCT2 antibody concentration. A 0.2 second exposure time showed the least background noise, while both 8 µg/mL and 12 µg/mL of anti-CCT2 antibody resulted in a detectable signal, S5 Fig. We chose 8 µg/mL as the anti-CCT2 antibody concentration moving forward.

## Detection of CCT2 positive tumor cells in blood

To investigate how CCT2 staining impacted CTC visualization using CSS Analyzer II, we used the method of spiking a known number of cancer cells into a vial of freshly collected blood from healthy individuals. This method allowed us to evaluate and validate the CTC detection protocol that incorporates the CCT2 signal under conditions of a consistent number of cancer cells, which is not possible using patient blood. The standard FDA-approved protocol for CTC detection uses the CSS CTC kit in which the signal for CK is read in the PE channel. For this study, since we wanted to do a comparative analysis, we used the CSS CXC kit in which the PE channel is available for customization of the CCT2 signal and CK is read in the FITC channel. After scanning the sample, the CSS Analyzer II creates "events" based on a positive CK signal near a positive DAPI signal. These "events" must then be assigned by an operator as a "CTC" or "not a CTC" based on the CXC kit criteria. This criterion includes: 1) a DAPI positive signal, 2) a CK-FITC positive signal that overlaps with 50% of the DAPI signal, and 3) a negative CD45 signal. A representative page from a CSS Analyzer II readout is shown in S6 Fig. The CSS software runs on Linux; therefore, the images appear to have low-quality pixelation. However, this specific pixelation level was approved by the FDA for the purposes of identifying CTCs and therefore cannot be changed by the manufacturer. Normal healthy human blood does not contain CTCs and therefore the machine picks up random signals of CK and DAPI that are near to each other. Examples of this are seen in Fig 4A, rows 1 and 2. It is also common to see leukocytes that are CK positive as seen in row 3. None of these cells in rows 1–3 are CTCs, based on the CXC kit definition. For this study, we spiked ~1,000 cells from either the MDA-MB-231 or the T47D-CCT2 cell lines into 10 mL of healthy human blood. An operator then sorted the sample through the CSS Analyzer II events to select spiked cancer cells based on 1) CTC criteria from the CXC kit and 2) positive signal in the PE channel (column five) for anti-CCT2 antibody staining intensity as described in the methods. Fig 4B and 4C, show that CCT2 was detected in spiked MDA-MB-231 and T47D-CCT2 cells with minimal background inference in the column five. Rows 4, 5, 9, and 10 show typical spiked cancer cells that were CK positive and CCT2 positive. Rows 6 and 7 show examples of spiked cancer cells that were CK dim, as indicated by the red arrows, but were CCT2 positive and CD45 negative. These non-leukocytic cells would have been missed by the classical definition for CTC. Images in Fig

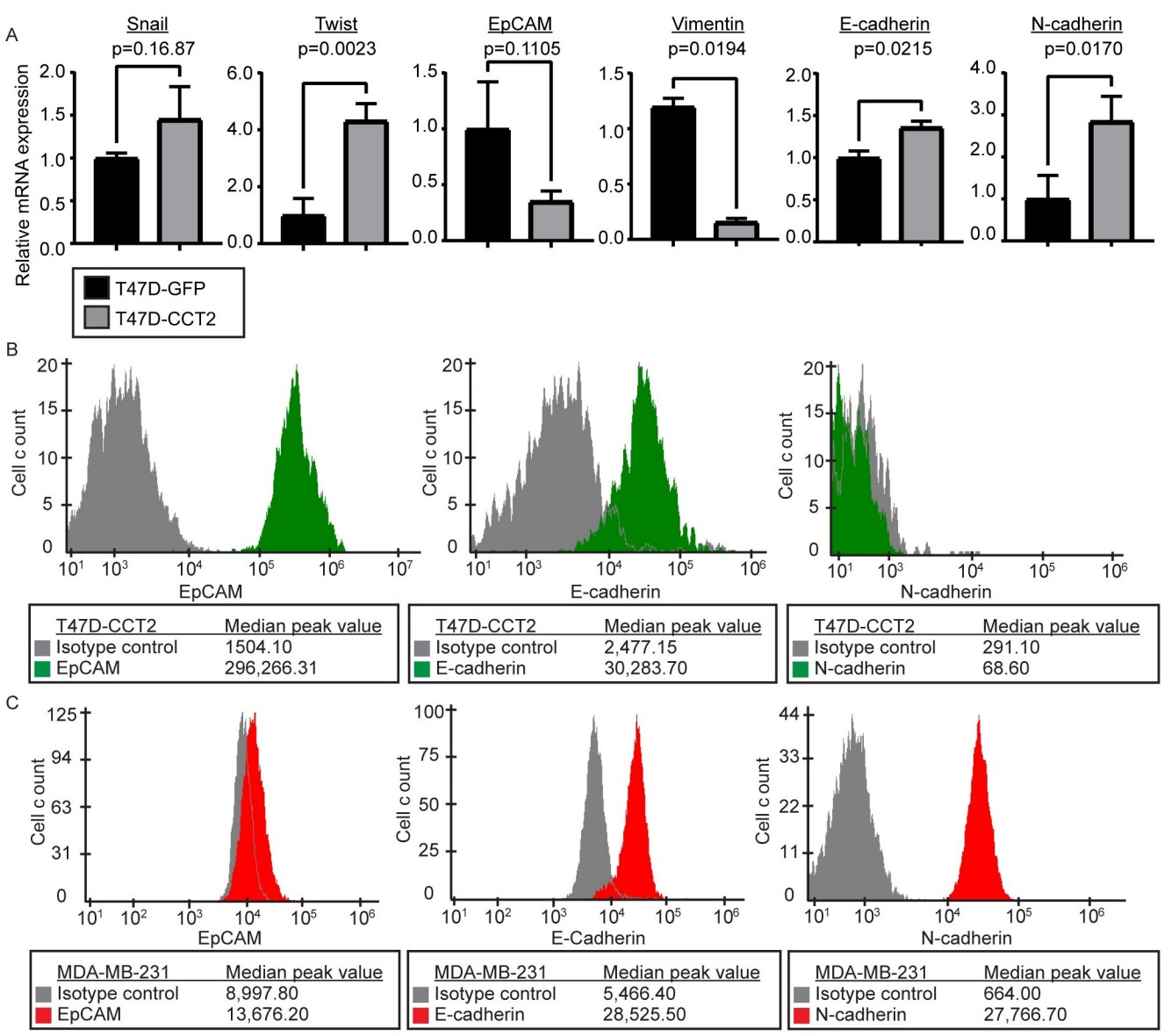

**Fig 3. EpCAM, vimentin, E-cadherin, and N-cadherin expression in MDA-MB-231 and T47D cells.** (A) RT-PCR data comparing lentiviral control (T47D-GFP) (black) with T47D-CCT2 (grey) for expression of EMT genetic markers: *SNAIL* and *TWIST*, epithelial markers: *EpCAM* and *E-cadherin*, and mesenchymal markers: *vimentin* and *N-cadherin*. p-values are shown on the graph. (B) Flow cytometry data detecting surface expression of EpCAM, E-cadherin, and N-cadherin protein in T47D-CCT2 cells (green) compared to isotype controls (grey). (C) Flow cytometry data detecting surface expression of EpCAM, E-cadherin, and N-cadherin proteins in MDA-MB-231 cells (red) compared to isotype controls (grey). All experiments were performed in duplicate.

4 also showed that spiked cancer cells have variations in CCT2 staining intensity. This is seen by the doublet in row 11, where the blue arrow points to a cell with a slightly lower CCT2 intensity than its partner in the same frame and by the cell in row 13 which has dim CCT2 staining, as shown by the grey arrow. The doublet finding is especially significant since it demonstrates that variations in the CCT staining intensity (dim vs bright) are detectable in the CSS or other similar CTC detection systems. Finally, it is important to note that while some leukocytes are CCT2 positive (light blue arrow in row 3), this is not always the case (yellow arrows in rows 8 and 12), and leukocytes are ruled out as cancer cells due to positive CD45 signal.

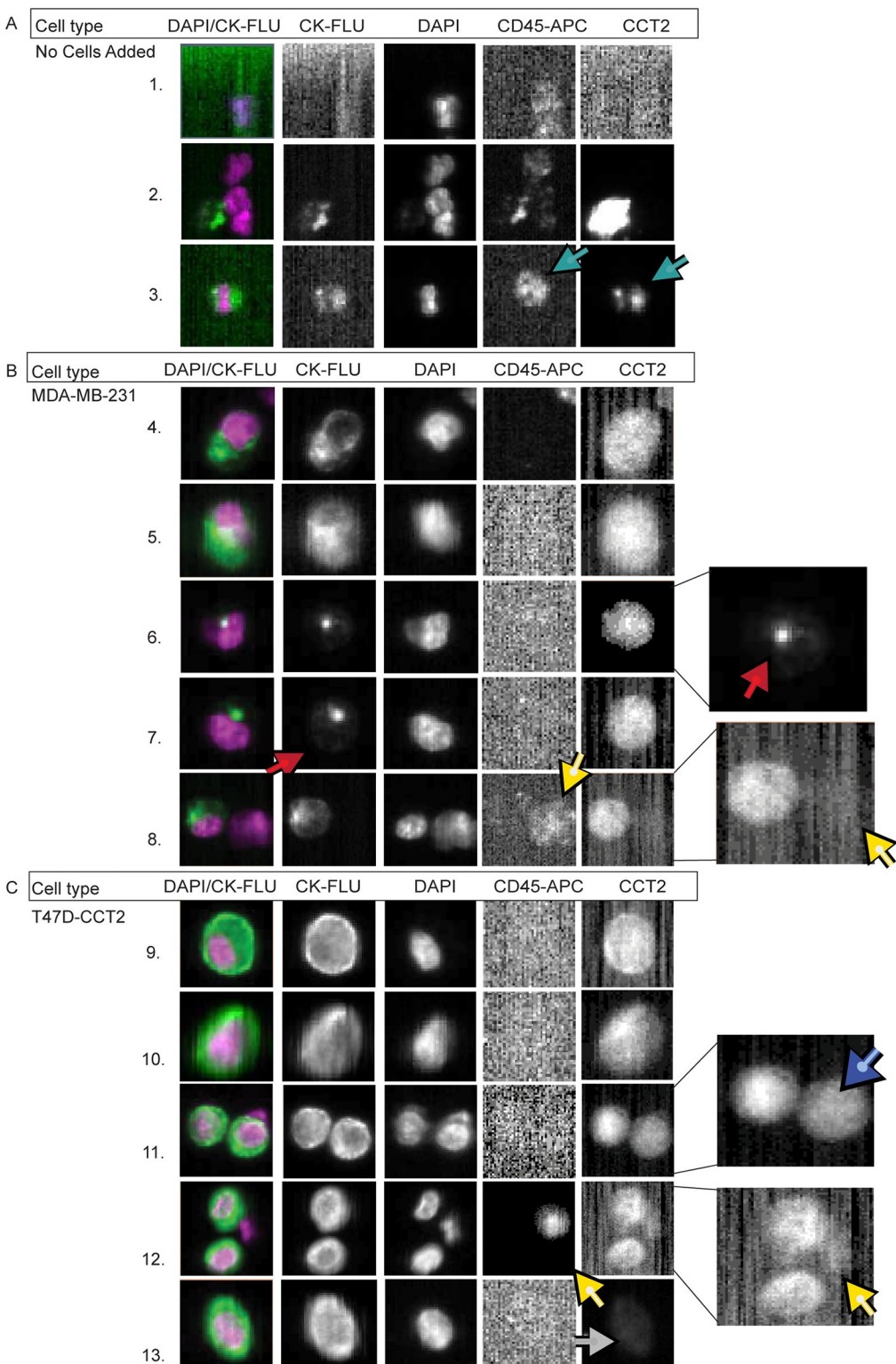

**Fig 4. Breast cancer cells spiked in blood can be detected based on CCT2 staining using the CSS.** Representative images of (A) whole healthy human blood without spiked cancer cells processed through the CSS and stained for CCT2, and (B-C) Representative images of MDA-MB-231 (B) or T47D-CCT2 (C) cells spiked into human blood, processed through the CSS, and stained for CCT2. Light blue arrows: leukocyte that is CCT2 positive. Red arrows: cells with dim CK signal and CCT2 positive signal. Yellow arrows: leukocytes that have dim CCT2 signal. Dark blue arrow: doublet of

spiked cancer cells with different CCT2 staining intensities. Grey arrows: cells with dim CCT2 signal. CK-FLU. This data is representative of ten experiments.

The goal of the CTC enumeration is to determine if patients meet a threshold for prognostic purposes. Therefore, instead of trying to capture every CTC in the blood, the current technology may underestimate the number of CTCs in a vial of blood. With such low prognostic thresholds of 3 or 5 CTCs per 7.5 mL of blood, ways to improve recovery and detection of CTCs would be beneficial. The standard CSS CTC kit is around 81% efficient at recovering cells spiked into CSS Autoprep tubes and immediately analyzed by CSS [32]. For this study, we used the CSS CXC kit, which is estimated to have a 10–18% lower recovery than the CSS CTC kit [31]. With this in mind, we decided to quantitatively evaluate the blood spiked cancer cell experiment after processing through the CSS. We found that cancer cells added to the CellSave vials for ~22 hours, resulted in about 30% of cancer cell death. This timeframe was chosen to resemble the processing of patient blood specimens. Using our modified CTC definition for CK positive and CCT2 positive cells, the average recovery was 62% for MDA-MB-231 cells and 64% for T47D-CCT2 cells (Table 4). This puts us to around 92–94% recovery when accounting for the loss of cell viability during storage overnight in the CellSave vial. Looking at the images from the CSS Analyzer II, the MDA-MB-231 cells more often showed multiple cells with dim CK fluorescence compared to the T47D-CCT2, which may be representative of reduced epithelial expression that was noted in Fig 3C and S3 Fig. Therefore, we analyzed the dim CK/CCT2 positive ($CK^-/CCT2^+$) population of cells and found that it included 5.6% of the MDA-MB-231 cells and 1.3% of the T47D-CCT2 cells. This suggests that CCT2 staining could improve the detection of cells like MDA-MB-231 that display mesenchymal features.

To confirm that the ($CK^-/CCT2^+$) cells detected in our analysis were the cancer cells that we originally spiked and not random blood cells from the donor, we stained for CD44, a tumor marker associated with MDA-MB-231 cells. Flow cytometry results showed that the recovered ($CK^-/CCT2^+$) cells also expressed CD44, Fig 5. These CD44 positive cells were absent in the blood-alone control lacking MDA-MB-231 cells, confirming the results from the CSS Analyzer II where the CCT2 stain was used to detect spiked cancer cells.

To determine if the detection of CCT2 in breast cancer cells could be extended to other cancer cells, we spiked CRL5853 and CRL5903 SCLC cell lines into blood and performed CCT2 staining in the CSS platform. Select images representing different recovered SCLC cells are shown in Fig 6A and 6B. Cells from both SCLC cell lines showed positive CCT2 staining with similar trends to what was seen in breast cancer. Typical spiked cancer cells that were both CK positive and CCT2 positive are seen in rows 1, 3, and 5. Red arrows designate ($CK^-/CCT2^+$) spiked cancer cells in rows 2 and 4. A doublet of cells with different staining intensities for

**Table 4. Recovery of breast cancer cells spiked in blood and analyzed with the CSS Analyzer II.**

| | PERCENT FROM CELL SPIKE: (1,000 CELLS/10 mL blood) | |
|---|---|---|
| *Cell Type* | *$CK^+$ cells* | *$CK^+$ and $CCT2^+$ cells* |
| MDA-MB-231 | 58.60% | 62.10% |
| T47D-CCT2 | 63.10% | 64.00% |
| | PERCENT FROM TOTAL CELL COUNT: CTC AND ($CK^-/CCT2^+$) | |
| *Cell Type* | *$CK^+/CCT2^-$ cells* | *$CK^-/CCT2^+$ cells* |
| MDA-MB-231 | 7.61% | 5.62% |
| T47D-CCT2 | 14.66% | 1.28% |

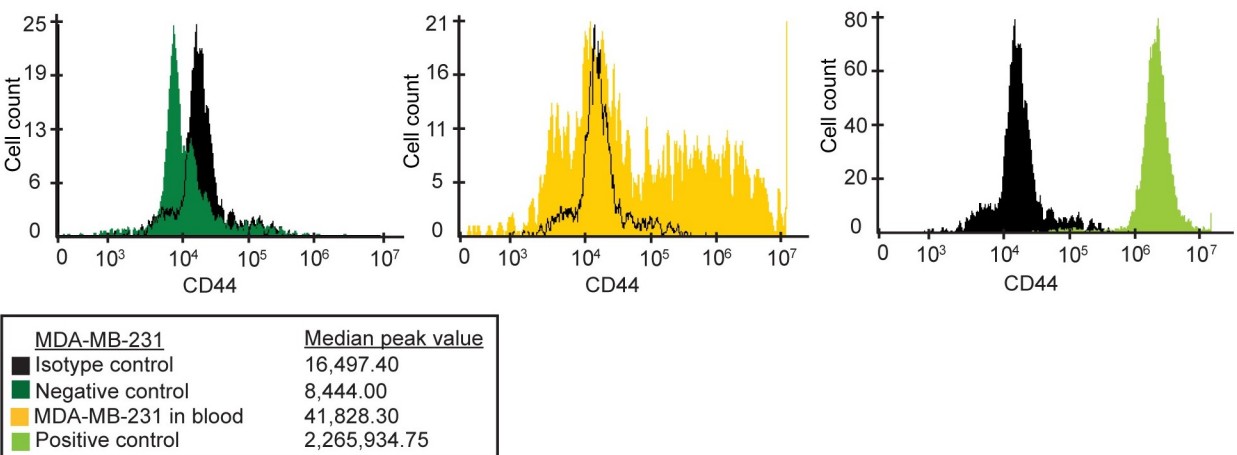

**Fig 5. Recovered breast cancer cells that have dim CK and are CCT2 positive, also express CD44.** MDA-MB-231 cells recovered from the spike in blood were stained and measured for CD44 levels in gated CCT2 positive cells. Alexa-405 served as the isotype control (black). The left panel shows the negative control which was healthy blood cells only stained for CD44 (dark green). The middle panel shows the CCT2 positive spiked cancer cells stained for CD44 (yellow). The right panel shows the positive control which was MDA-MB-231 cells (not spiked in blood) stained for CD44 (light green).

CCT2 is shown in row 5 (blue arrow). We also tested lower anti-CCT2 antibody concentrations in the spiked lung cancer cells and still saw strong staining, Fig 6C–6E. Most notably, several cells had dim CK staining or had CK that was blocked by DAPI staining while remaining CCT2 positive (rows 7, 9, and 11). These results demonstrate that intracellular staining of CCT2 may have broad application in the detection of shed tumor cells in blood in other cancers.

## CCT2 staining of CTC tested with blood from SCLC patients

Having shown that CCT2 staining works in a standard CTC detection protocol using cancer cells spiked into blood, we performed a small pilot study using blood from SCLC patients to determine if CCT2 staining could detect CTCs released into blood in these patients. We chose SCLC, since such patients typically have greater number of CTCs compared to other cancer types [33]. The results confirm that CCT2 staining can be observed in CTCs detected in SCLC patient blood and that a range of detection is possible based on the saturating amounts on anti-CCT2 antibody used, Fig 7A. Three patterns of CCT2 staining of CTCs were noted. In patient 1, no CTCs based on standard criteria (>50% overlap of DAPI and CK staining, while CD45 negative) were noted, but dim CCT2 staining in some cells was observed; in patient 2, numerous CTCs were found that had dim or negative CCT2 staining; and patients 3 and 4, most CTCs detected were both CK positive and CCT2 positive, Fig 7B. To determine whether CCT2 could be used to detect CTCs in place of CK, we used a different algorithm for analysis of CTC from the CSS analyzer, based on a >50% overlap of DAPI and CCT2, while CD45 negative. We detected many more CTCs that were CCT2 positive and CK negative in all four patient samples, Fig 7C. These results indicate that the use of CCT2 as a marker for CTCs is feasible with the current CTC technology, as exemplified by the CSS, and that in cancers like SCLC, could provide an additional discriminatory marker to improve enumeration as well as provide biological data that could be relevant to the invasive nature of the CTCs being detected.

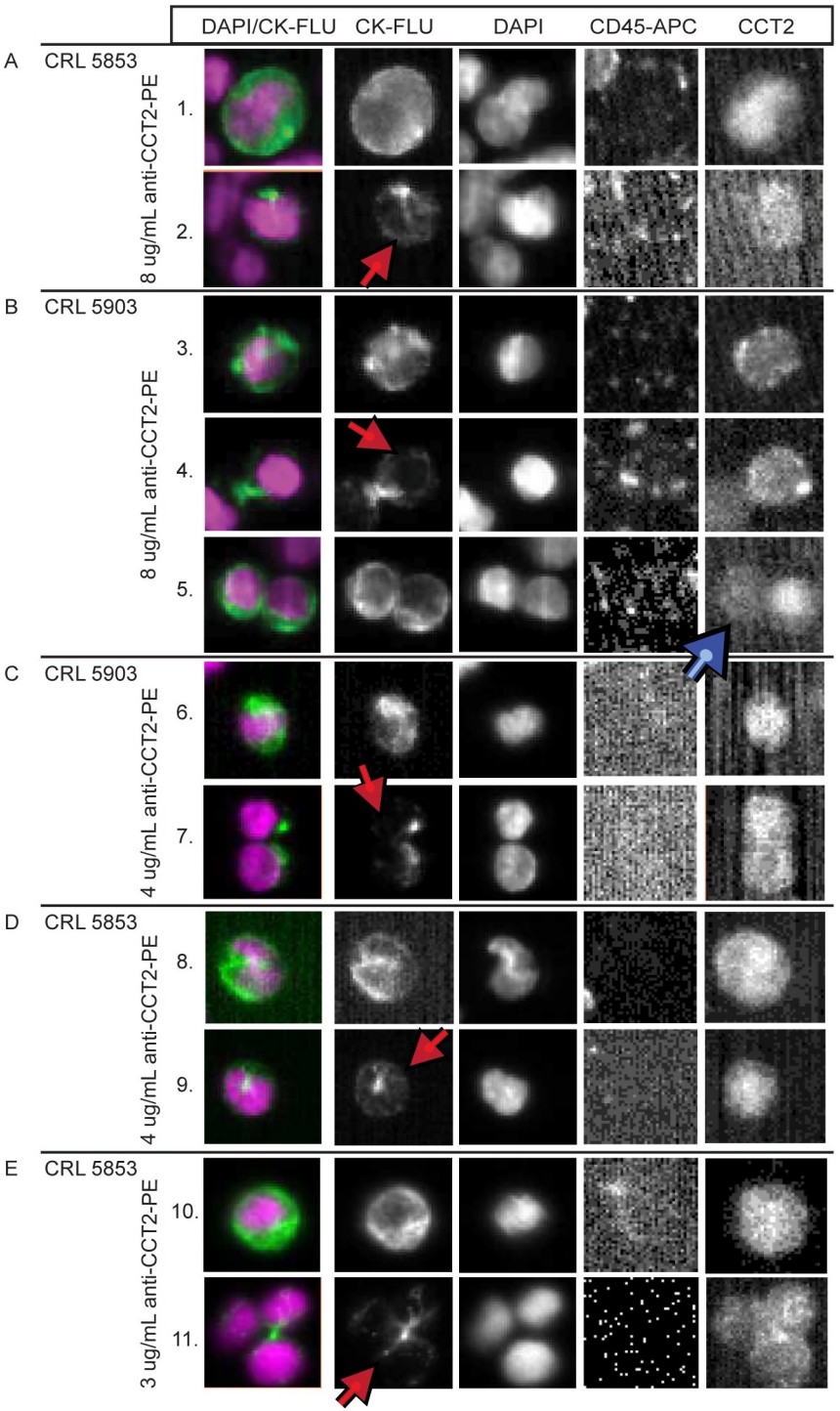

**Fig 6. CCT2 staining in lung cancer cell lines.** (A-B) Representative images from CSS Analyzer II show SCLC cells spiked into healthy human blood and processed with the CSS Autoprep using the anti-CCT2-PE antibody. (A) CRL 5853 and (B) CRL 5903. Red arrows: cells that have dim CK and are CCT2 positive. Blue arrows: doublet of spiked cancer cells with different CCT2 staining intensities. The experiment was performed in duplicate. (C-E) Representative images of lung cancer cells (C) CRL 5903 and (D-E) CRL 5853, stained with reduced anti-CCT2 antibody as indicated in Methods. Red arrows: cells with dim CK staining and CCT2 positive staining. CK-FLU.

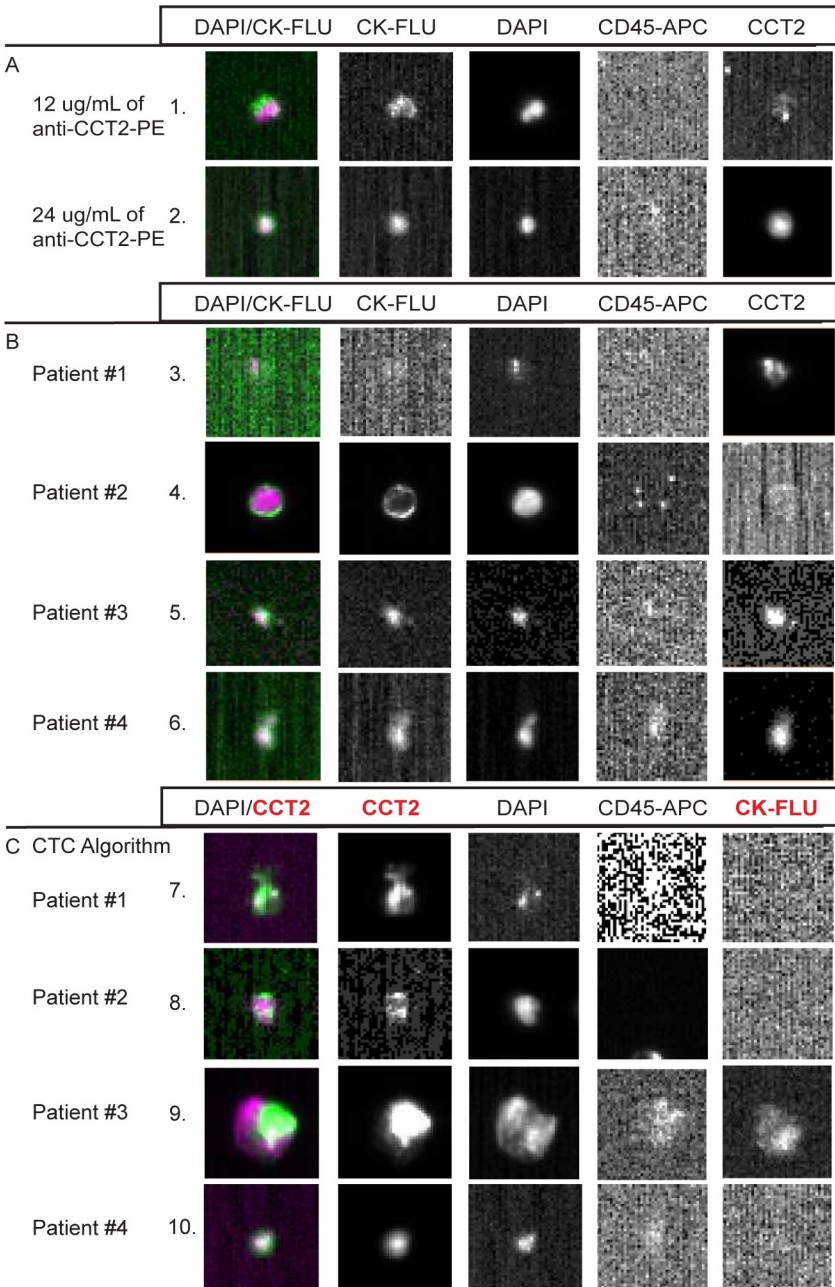

**Fig 7. Representative images from SCLC patient CTCs stained for CCT2.** (A) Representative images of CTCs, based on standard CTC criteria that were CCT2 positive at varying concentrations of the anti-CCT2-PE antibody. (B) Representative images of CTCs from each SCLC patient. Each image was taken from collections of relevant events that were analyzed using standard CTC criteria for the CSS CXC kits as described above. (C) Representative images from CTCs collected using the CTC analysis algorithm (instead of CXC analysis algorithm) where the DAPI signal overlaps with CCT2-PE instead of CK-FLU as in (A, B). Note that images that contain faint CD45 expression are a result of bleed-over from the signal in the PE channel.

## Discussion

In this study, we report that CCT2 expression is increased in primary cancer and metastatic tissue compared to normal tissue, and this inversely correlated with patient OS. CCT2 protein

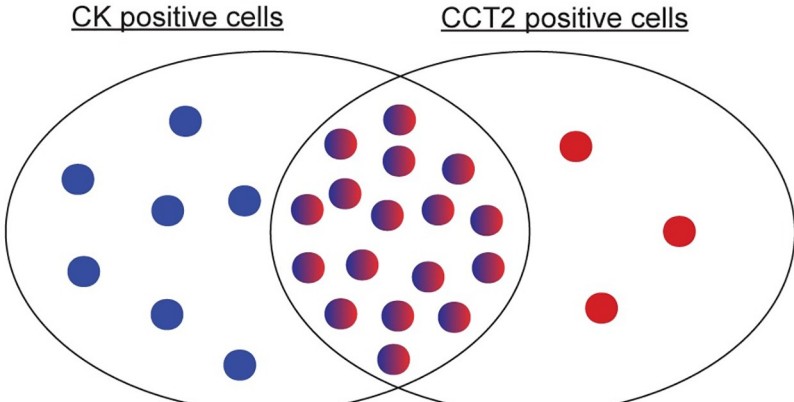

**Fig 8. Venn diagram showing possible CTC subsets that are CK⁺/CCT2⁻, CK⁺/CCT2⁺, and CK⁻/CCT2⁺.**

levels were minimal in normal tissues and increased in cancerous tissue. In an MBC patient cohort, CCT2$^{hi}$ tumor scores correlated inversely with recent monitoring indicative of active disease. The addition of CCT2 staining to the CSS led to improved recovery of cancer cells spiked into blood, including those with mesenchymal features, providing visual differences that could help classify CTCs. This was most notable in the detection of breast and lung cancer cells spiked into blood that were CK dim but CCT2 positive and not leukocytes. Using the modified CTC definition (incorporating CCT2 in addition to epithelial markers), our data revealed three possible CTC populations; CK⁺/CCT2⁻; CK⁺/CCT2⁺; CK⁻/CCT2⁺ as shown in the Venn diagram in Fig 8. Current CTC detection methods recover cells from the first two populations and have found prognostic clinical significance in CTC counts above certain thresholds in breast, colon, and prostate cancers. The bioinformatics and histological data support that the third population of cells (CK⁻/CCT2⁺) may have similar clinical significance, due to CCT2's association with metastatic tissue and patient OS and was observed in cells derived from MBC and SCLC lines spiked into blood as well as CTCs from SCLC patients.

While most CTC enrichment technologies focus on either physical or biological properties, staining for CCT2 to identify CTCs could have a wide application that is particularly beneficial for tumor heterogeneity. In our study, we used the CSS platform as a starting point to evaluate CCT2 as a marker for CTCs because it is an FDA-approved CTC enumeration technique whose use could advance future clinical applications. The CSS employs EpCAM-based isolation methods and therefore may miss EpCAM negative CTCs as these undergo EMT. CSS addresses this concern with a capture enhancement reagent that enables controlled EpCAM-ferrofluid nanoparticle aggregation so that even cells with low EpCAM levels, like the MDA-MB-231 cells we used (S3 Fig), are recovered. Additionally, some studies found no correlation between CTCs with low EpCAM expression and patient OS [34, 35]. Nevertheless, it is a limitation in our study that any EpCAM negative and CCT2 positive cells potentially could have been missed.

Post-CTC enrichment, the CSS uses the epithelial marker CK for staining and CTC visualization and enumeration. Our bioinformatics data shows that CK levels are low in metastatic tissue, which suggests that using CK for CTC discrimination may overlook more invasive-prone CTCs. Additionally, studies show that while CK 8, 18, and 19 are present in luminal breast cancers, basal breast cancers have a stronger correlation with CK 5, 6, and 14 [36–38]. Researchers have started addressing this concern by adapting CSS with additional antibody markers, including vimentin [39], HER2 [40], and CD44 [41], among others [42]. However,

none of these have been adapted for clinical use. Some of the challenges using these markers are that they cannot account for heterogeneous tumors or different cancer types, and can be involved in hematopoietic processes, which complicates differentiation from blood cells. These markers are also membrane or surface proteins. CCT2 is an intracellular protein that may not be subject to the variability of plasma membrane turnover or shedding and is highly expressed in multiple cancers, giving it a potentially broader application base [19]. Our lab previously showed that CCT2 expression did not correlate with ER or PR status and therefore could be used with different types of breast cancer including TNBC [12]. This demonstrates that CCT2 as a biomarker could bring a unique solution to the problem of tumor heterogeneity.

In the future, CTC detection will remove operator error by automating the process. An example of this is the open software ACCEPT (Automated CTC Classification Enumeration and PhenoTyping) [43]. This open access software allows the operator to create criteria (such as cell roundness or intensity of CK stain) for automated CTC detection [44]. In a study using ACCEPT, various populations of EpCAM-enriched circulating cells in non-small cell lung cancer (NSCLC) patient blood and healthy blood with spiked cancer cells were analyzed by adding a stain for a granulocyte marker (such as CD16), allowing them to decrease the number of unidentified nucleated cells in the blood [44]. Such work aligns with our own study of CCT2 to improve the identification of cancer cells in blood and detect the most clinically relevant CTCs for monitoring patient outcomes. Additionally, ACCEPT measures stain intensity which allows for operators to create thresholds, such as for the breast cancer therapeutic marker: HER-2 [43]. Addition of ACCEPT to operator classifications for HER-2 positive cells increased recognition from 30% to 51%, which allowed for a more reproducible way to guide treatment options in breast cancer patients [43]. We envision that CCT2 could be adapted in a similar manner for automated assessment of CTCs in multiple cancers.

Staining for CCT2 or other chaperones can go beyond CTCs. One study used chaperones for the enrichment of CTCs in head and neck cancer or lung cancer patients [45]. Since then, a clinical trial has been recruiting patients with various metastatic cancers to measure CTC levels when enriched with membranous heat shock protein (mHSP70) vs EpCAM (NCT04628806). This study supports our findings that a marker like CCT2 whose expression correlates with advanced cancers could have applications for enrichment and detection of metastatic cancers that have either epithelial or mesenchymal characteristics. Several liquid biopsy tissues are under study, including ctDNA, circulating hybrid cells [46], and extracellular vesicles (EVs). In a review that discussed extracellular HSP and cancer, they retrospectively evaluated mass spectrometry data of EVs from studies that had clinical cancer patient samples to measure the expression of HSP and their co-chaperones in the EVs [47]. Their analysis revealed the expression of CCT2 and the other CCT subunits in the EVs of several cancers. A more recent review of EVs included a discussion of CCT in EVs and cancer and their role providing diagnostic and prognostic information [48]. Increased expression of CCT2, CCT6A, and CCT7 in EVs from cavitron ultrasonic surgical aspirators (CUSA) samples in glioblastoma patients correlated with poor patient prognosis. This study also observed increased copy number of all *CCT* subunits, except *CCT1*, that correlated with poor patient outcomes [49]. Such reports support our findings that CCT2 detection has potential applications in liquid biopsies approaches and could provide biologically relevant information to advance a minimal invasive option for monitoring tumor metastatic potential and progression.

## Supporting information

**S1 Fig. UCSC Xena database comparing all eight CCT subunits in normal (GTEx) vs. cancerous (TCGA) tissue.** (A) Overall cancer (n = 17,200): the most significant differences

were observed in the *CCT2* and *CCT3* genes, and the least difference was in the *CCT6B* gene. (B) Brain cancer (n = 1,277): the most significant difference was seen in the *CCT2* gene, and the least difference was in the *CCT6B* gene. (C) Breast cancer (n = 1,660): the most significant differences were seen in the *CCT3* and *CCT8* genes and the least difference was in the *CCT6B* gene. (D) Colon cancer (n = 598): the most significant difference was seen in the *CCT2* gene, and the least difference was in the *CCT6B* gene. (E) Lung cancer (n = 1,301): the most significant differences were seen in the *CCT2* and *CCT3* genes, and the least difference was in the *CCT6B* gene. p<0.0001 for all samples.
(TIF)

**S2 Fig. CCT2 score scale and histological analysis of CCT2 and STAT3 in cancerous breast tissue.** (A) Representative images of breast cancer tissue at each CCT2 stain score 0–4 with an explanation for the stain score scale. Images are from 0: TMABRN801b sample E1, 1: patient 25, 2: patient 08, 3: patient 33, 4: patient 37. (B-C) Breast cancer TMA Core Matrix from Florida Hospital stained for (B) CCT2 and (C) STAT3.
(TIF)

**S3 Fig. Broad Institute Cancer Cell Line Encyclopedia (CCLE) RNAseq expression data.** Breast cancer cell lines (purple dots) with a focus on T47D (green dot) and MDA-MB-231 (red dot) cell lines for *CDH1* (E-cadherin) vs. *CDH2* (N-cadherin), and *EpCAM* vs *VIM* (vimentin).
(TIF)

**S4 Fig. Anti-CCT2-PE validated with PE-isotype control under different conditions.** (A-B) Tested anti-CCT2-PE concentrations: 0 μg/ml (black), 5 μg/mL (blue), 10 μg/mL (purple), 20 μg/mL (green), and 25 μg/mL (yellow) in (A) MDA-MB-231 and (B) T47D-CCT2 cell lines with PE-isotype control (grey). (C) Tested various concentrations of MDA-MB-231 cells as indicated, with 10 μg/mL of anti-CCT2-PE (black/dark red/dark green) and PE-isotype control (light grey/red/green). (D) Tested 5X concentration of PE-isotype control (grey) with 70 min incubation in MDA-MD-231 cells with anti-CCT2-PE concentrations: 0 μg/mL (black), 5.5 μg/mL (purple), 11.0 μg/mL (green), 16.5 μg/mL (yellow), 22.0 μg/mL (blue), and 33.0 μg/mL (red).
(TIF)

**S5 Fig. Anti-CCT2-PE exposure time and antibody concentration tested in CellSearch® System (CSS) Analyzer II.** (A) Representative images from CSS Analyzer II of spiked cancer cells stained with 0 μg/mL anti-CCT2-PE at three different exposure times: 0.2, 0.3, and 0.4 sec. (B) Images from CSS Analyzer II of spiked cancer cells at 0.2 seconds exposure stained with three different concentrations of anti-CCT2-PE: 4 μg/mL, 8 μg/mL, and 12 μg/mL as indicated. Column one: overlay of columns two and three. Column two: CK-FLU-FITC signal. Column three: DAPI signal. Column four: CD45-APC signal. Column five: this column was used to stain for anti-CCT2-PE. The experiment was completed in duplicate.
(TIF)

**S6 Fig. Representative page from CellSearch (CSS) Analyzer II.** The orange box indicates an event that was selected by the operator as a spiked cancer cell or CCT2 positive cell. Column one: overlay of columns two and three. Column two: CK-FLU-FITC signal. Column three: DAPI signal. Column four: CD45-APC signal. Column five: this column was used to stain for anti-CCT2-PE. This includes representative images from ten experiments.
(TIF)

**S1 Table. Primer sequences for a reverse transcription-polymerase chain reaction.**
(TIF)

**S2 Table. Statistics comparing CCT2, KRT8, KRT18, and KRT19 gene expression in TCGA and GTEx samples.**
(TIF)

**S3 Table. TMA core matrix tumor score results for CCT2 and STAT3.**
(TIF)

**S4 Table. Metastatic breast cancer patient information.** NG: information not given, Y: yes, N: no.
(TIF)

## Acknowledgments

The authors are grateful to all participating patients and their families.

## Author Contributions

**Conceptualization:** Amanda Cox, Ana Martini, Annette R. Khaled.

**Data curation:** Amanda Cox, Ana Martini, Xiang Zhu, Eunkyung Lee, Amr S. Khaled, Louis Barr, Carlos Alemany, Na'im Fanaian, Elizabeth Griffith, Ryan Sause, S. A. Litherland, Annette R. Khaled.

**Formal analysis:** Heba Ghozlan, Xiang Zhu, Eunkyung Lee, Amr S. Khaled, Annette R. Khaled.

**Funding acquisition:** Annette R. Khaled.

**Investigation:** Amanda Cox, Ana Martini, Eunkyung Lee, S. A. Litherland, Annette R. Khaled.

**Methodology:** Amanda Cox, Ana Martini, Rebecca Moroose, Xiang Zhu, Amr S. Khaled, Annette R. Khaled.

**Project administration:** Rebecca Moroose, Annette R. Khaled.

**Resources:** Rebecca Moroose, Eunkyung Lee, Carlos Alemany, Elizabeth Griffith, Ryan Sause, S. A. Litherland, Annette R. Khaled.

**Supervision:** Annette R. Khaled.

**Validation:** Amanda Cox, Heba Ghozlan, Xiang Zhu, Annette R. Khaled.

**Visualization:** Amanda Cox.

**Writing – original draft:** Amanda Cox.

**Writing – review & editing:** Ana Martini, Heba Ghozlan, Rebecca Moroose, Xiang Zhu, Eunkyung Lee, Amr S. Khaled, Louis Barr, Carlos Alemany, Na'im Fanaian, S. A. Litherland, Annette R. Khaled.

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
