## [Decision Letter · Decision Letter 0]

12 Oct 2021

PONE-D-21-29407Chaperonin Containing TCP1 as an actionable marker for detection of rare cancer cells shed in bloodPLOS ONE

Dear Dr. Khaled,

Thank you for submitting your manuscript to PLOS ONE. After careful consideration, we feel that it has merit but does not fully meet PLOS ONE’s publication criteria as it currently stands. Therefore, we invite you to submit a revised version of the manuscript that addresses the points raised during the review process. Based on the evaluation of the manuscript and on the comments from reviewers, the topic could be of interest for the readers but the manuscript would need a significant improvement in terms of study design to be effectively considered for publication. In particular, to demonstrate the value of the proposed method to detect circulating cells from blood authors should employ blood samples from cancer patients instead of using just healthy subjects (see below the comments). Thus, if authors will address these major concerns the paper will be appropriately re-evaluated for publication.

We look forward to receiving your revised manuscript.

Kind regards,

Vincenzo L'Imperio

Academic Editor

PLOS ONE

Journal Requirements:

"I have read the journal's policy and one of the authors of this manuscript (Dr. Annette Khaled) has the following competing interests: [shareholder in Seva Therapeutics, Inc.] "

We note that one or more of the authors are employed by a commercial company: Seva Therapeutics, Inc.

“The funder provided support in the form of salaries for authors AK, but did not have any additional role in the study design, data collection and analysis, decision to publish, or preparation of the manuscript. The specific roles of these authors are articulated in the ‘author contributions’ section.”

Please respond by return email with an updated Funding Statement and Competing Interests Statement and we will change the online submission form on your behalf.

Reviewers' comments:

Reviewer's Responses to Questions

**Comments to the Author**

1. Is the manuscript technically sound, and do the data support the conclusions?

Reviewer #1: Yes

Reviewer #2: No

2. Has the statistical analysis been performed appropriately and rigorously? 

Reviewer #1: Yes

Reviewer #2: I Don't Know

3. Have the authors made all data underlying the findings in their manuscript fully available?

Reviewer #1: Yes

Reviewer #2: Yes

4. Is the manuscript presented in an intelligible fashion and written in standard English?

Reviewer #1: Yes

Reviewer #2: Yes

5. Review Comments to the Author

Reviewer #1: The manuscript entitled "Chaperonin Containing TCP1 as an actionable marker for detection of rare cancer cells shed in blood" highlighted that detection of CCT2 could identify tumor cells shed into blood and has application in liquid biopsy approaches, such as detection of CTCs, to enable the use of minimally invasive methods for cancer diagnosis.

The manuscript is really well written and of interest for the readers.

Minor comments:

- The Authors should provide the expand forms for all acronyms, including gene acronyms, thorugh the text when they first appear.

- Gene acronyms should be written in italics.

- The Authors should provide high quality figures.

Reviewer #2: Dear authors thanks for providing your manuscript,

Main comment:

Authors evaluated the use of intracellular CCT2 staining for visualization of breast cancer cells spiked into blood from healthy donors. "This approach served as a surrogate for cancer patient blood."

If i did not misunderstand, authors did not use the patients blood sample. I would strongly suggest to have used the patients blood samples and the healthy donors as controls not as a surrogate.

-Please change the topic of this manuscript with a more specific one as its not clearly informative.

-the introduction and the method section is too long and difficult to follow. the text should be more into point and straight forward. also some subheadings could help (method section).

-please clarify why the authors did not use the patients blood sample.

---

## [Author Response · Author response to Decision Letter 0]

22 Dec 2021

The editor requested that we demonstrate the value of using blood from healthily subjects. We explained that the blood from healthy subjects was spiked with cancer cells and used to develop the novel of use of the chaperonin to detect cancer cells in blood. Normally, healthy blood does not contain any cancer cells. So any cancer cells detected in spiked normal blood would be the spiked cancer cells. This approach is commonly used in multiple publications to test methods for detecting cancer cells in blood. We referenced three of these publications in our response to reviewers. In addition, we also purchased blood from a small number of cancer patients to confirm that the findings made with cancer cell spiked blood was also achieved using patient blood. This new data has been added to the manuscript.

---

## [Decision Letter · Decision Letter 1]

15 Feb 2022

Chaperonin Containing TCP1 as a marker for identification of circulating tumor cells in blood

PONE-D-21-29407R1

Dear Dr. Khaled,

We’re pleased to inform you that your manuscript has been judged scientifically suitable for publication and will be formally accepted for publication once it meets all outstanding technical requirements.

Kind regards,

Vincenzo L'Imperio

Academic Editor

PLOS ONE

Reviewers' comments:

Reviewer's Responses to Questions

**Comments to the Author**

1. If the authors have adequately addressed your comments raised in a previous round of review and you feel that this manuscript is now acceptable for publication, you may indicate that here to bypass the “Comments to the Author” section, enter your conflict of interest statement in the “Confidential to Editor” section, and submit your "Accept" recommendation.

Reviewer #1: All comments have been addressed

Reviewer #2: All comments have been addressed

2. Is the manuscript technically sound, and do the data support the conclusions?

Reviewer #1: Yes

Reviewer #2: Yes

3. Has the statistical analysis been performed appropriately and rigorously? 

Reviewer #1: Yes

Reviewer #2: I Don't Know

4. Have the authors made all data underlying the findings in their manuscript fully available?

Reviewer #1: Yes

Reviewer #2: Yes

5. Is the manuscript presented in an intelligible fashion and written in standard English?

Reviewer #1: Yes

Reviewer #2: Yes

6. Review Comments to the Author

Reviewer #1: In the recised Version of the paper the Authors have addressed all my concerns and I have no further comments.

Reviewer #2: The manuscript seems to be in a good status. The comments have been addressed. The figures are not of a high quality however i bet that later on the authors will submit high quality ones.

7. PLOS authors have the option to publish the peer review history of their article (what does this mean?). If published, this will include your full peer review and any attached files.

Reviewer #1: No

Reviewer #2: No

---

## [Editor Report · Acceptance letter]

28 Feb 2022

PONE-D-21-29407R1 

Chaperonin Containing TCP1 as a marker for identification of circulating tumor cells in blood 

Dear Dr. Khaled:

I'm pleased to inform you that your manuscript has been deemed suitable for publication in PLOS ONE. Congratulations! Your manuscript is now with our production department. 

Kind regards, 

on behalf of

Dr. Vincenzo L'Imperio 

Academic Editor

PLOS ONE